# Text-Guided In-Context Forecasting for Multimodal Time Series Prediction

## Abstract

Real-world time series forecasting often contains both numerical histories and textual context, such as news, reports, or event descriptions. Existing multimodal forecasting methods typically incorporate text through LLM-based prompting or representation-level fusion, which presents a modelling challenge when textual signals are heterogeneous, noisy and unevenly informative. We propose TAG-ICF, a text-attention guided in-context learning (ICL) forecasting framework that uses a frozen, pretrained ICL backbone for numerical prediction while using text to modulate attention over historical examples. Textual inputs are analyzed using LLMs and embedded as lagged features to compute query-context relevance scores, serving as an attention gate that prioritizes contextually relevant historical patterns. TAG-ICF consistently outperforms all multimodal baselines across nine datasets, yielding superior average ranks of 1.50 and 1.22 in MSE and MAE, respectively. Our framework surpasses numerical-only ICL on nearly all datasets and delivers up to 23.94% MAE improvement relative to the strongest competing methods in diverse settings.

## 1. Introduction

Time series forecasting is a cornerstone of decision-making in financial markets, energy systems, and macroeconomics (Liu et al., 2024; Williams et al., 2025). In these domains, predictive signals are inherently multimodal: numerical data captures endogenous historical dynamics, while unstructured text, such as news reports, policy briefings, or event descriptions, provides crucial context on exogenous drivers and regime shifts that are unobservable from numerical histories alone.

Existing multimodal forecasting approaches incorporate

textual information via LLM-based prompting (Xie et al., 2023; Yu et al., 2023; Wang et al., 2024) or embedding fusion (Jin et al., 2024; Sun et al., 2024; Jia et al., 2024; Cheng et al., 2025; Liu et al., 2025; Tang et al., 2025). Prompting-based methods leverage the zero-shot reasoning of pretrained LLMs over temporal sequences; however, LLMs often remain unreliable for precise numerical reasoning (Tan et al., 2024). Fusion-based methods, on the other hand, integrate textual representations with numerical features through embedding fusion or alignment. While effective in many multimodal settings, these approaches often require the model to learn not only the numerical forecasting structure, but also when and how textual context should influence the prediction. This challenge is pronounced in realistic settings where the textual context is heterogeneous and unevenly informative (Zhang et al., 2025). However, texts can also be useful in identifying historical situations, such as specific market shocks or events that are relevant to the current period, in contrast to numerical histories alone, which may fail to fully capture underlying temporal shifts. This insight aligns naturally with the In-Context Learning (ICL) framework, which leverages historical examples as context to perform prediction during inference. Such alignment enables the use of text as contextual guidance to modulate relevant historical patterns, bypassing the complexities of traditional representation-level fusion.

Recent ICL-based tabular foundation models, such as Tab-DPT (Ma et al., 2025) and TabPFN (Hollmann et al., 2023; 2025), have demonstrated strong performance on structured prediction tasks under this paradigm. Recent work has further extended ICL to time series forecasting, with approaches such as TabPFN-TS (Hoo et al., 2025) showing that structured in-context learning can be effective for sequential prediction. Separately, methods such as Context-Tab (Spinaci et al., 2025) explore incorporating textual information using representation-level fusion into ICL-based models for tabular prediction. However, multimodal time series forecasting with textual context under an ICL framework remains largely underexplored.

We propose TAG-ICF, a Text-Attention Guided In-Context Forecasting framework for multimodal numerical time series prediction. Our framework uses a frozen, pretrained ICL forecasting backbone to model temporal numerical structure, while textual context modulates the backbone's attention

[1]Anonymous Institution, Anonymous City, Anonymous Region, Anonymous Country. Correspondence to: Anonymous Author <anon.email@domain.com>.

over historical examples. Specifically, textual inputs are analyzed using LLMs, embedded, and organized into lagged textual representations, which are then used to compute context relevance scores between the query and candidate in-context examples. These relevance scores act as an attention gate, allowing text to guide the model toward historically relevant contexts without requiring representation-level fusion between text and numerical features.

We evaluate TAG-ICF on diverse multimodal benchmarks over 9 datasets spanning diverse domains. Our method outperforms the numerical-only ICL by 6.79% in MAE and 21.96% in MSE on average, consistently surpassing multimodal baselines with top average ranks of 1.50 (MSE) and 1.22 (MAE). TAG-ICF demonstrates the most stable and significant gains from textual integration, improving MAE by up to 23.94% relative to the strongest baseline while maintaining robustness across noisy, heterogeneous datasets.

## 2. Methodology

We introduce Text-Guided In-Context Forecasting (TAG-ICF), illustrated in Figure 1, a multimodal framework for time series prediction in which textual context modulates the relevance of historical examples within an in-context learning (ICL) framework. We instantiate this framework using TabDPT (Ma et al., 2025) for its efficiency, the mechanism however remains applicable to any transformer-based ICL backbone with accessible internal attention heads.

### 2.1. Problem formulation

Given a target series $\mathbf{y}_{1:T} = [y_1, y_2, \ldots, y_T]^\top$ ($y_t \in \mathbb{R}$), we aim to predict values for a horizon $H$, denoted as $\mathbf{y}_{T+1:T+H}$. Each historical step $t$ is associated with a multimodal context $\{t, \mathbf{x}_t, \mathbf{s}_t^{\text{raw}}, y_t\}$, where $\mathbf{x}_t \in \mathbb{R}^{d_n}$ and $\mathbf{s}_t^{\text{raw}}$ represent numerical and natural language covariates, respectively. Following the In-Context Learning (ICL) paradigm, we treat forecasting as conditional generation. While numerical and textual covariates $\{\mathbf{x}_t, \mathbf{s}_t^{\text{raw}}\}$ are unavailable for the prediction horizon $t \in \{T+1, \ldots, T+H\}$, we assume temporal features derived from step $t$, $\boldsymbol{\tau}_t \in \mathbb{R}^{d_\tau}$ are accessible for all time steps. The model generates the prediction as:

$$\hat{\mathbf{y}}_{T+1:T+H} = \text{TAG-ICF}(\{\boldsymbol{\tau}_t\}_{t=T+1}^{T+H} \mid \{\mathbf{x}_t, \mathbf{s}_t^{\text{raw}}, \boldsymbol{\tau}_t, y_t\}_{t=1}^{T}) \tag{1}$$

Performance is assessed over an evaluation period $\mathbf{y}_{T+1:T_{\text{eval}}}$, using a rolling-window approach. At each step $k \in [0, \ldots T_{\text{eval}-H}]$, the model uses the sequence up to $T+k$ as context to predict the subsequent horizon $\hat{\mathbf{y}}_{T+k+1:T+k+H}$.

### 2.2. Time series adaptations for ICL

To align multimodal observations with the ICL input paradigm, we first detail modality-specific adaptations for

numerical, timestamp, and textual data that preserve temporal dependencies. We then describe the task adaptation process, which structures historical data into context examples of lagged representations and future target values.

**Numerical representation:** Given numerical covariates $\mathbf{x}_t$ and target $y_t$, we construct a lagged feature vector in $\mathbb{R}^{L_{\text{num}} \cdot d_n + L_y}$ by concatenating them over lookback windows of length $L_{\text{num}}$ and $L_y$. Following Hoo et al. (2025), we augment this with $d_\tau$ timestamp features $\boldsymbol{\tau}_t$, including a monotonic running index, cyclical calendar attributes (e.g., hour, day, month, etc.) encoded via sine and cosine transformations, and $m$ seasonal features extracted via Fourier Transforms to capture domain-specific periodicities of target variable. These features enable temporal modeling within a tabular framework, resulting in the final numerical representation, denoted by $\mathbf{c}_t \in \mathbb{R}^{L_{\text{num}} \cdot d_n + L_y + d_\tau}$.

**Textual representation:** Raw textual data $\mathbf{s}_t^{\text{raw}}$ is often verbose and noisy. We use an LLM to synthesize high-level insights relevant to temporal dynamics, which are then encoded into $d_t$-dimensional embeddings using a pre-trained embedding model. By applying a consistent look-back window $L_{\text{text}}$, we construct the textual context matrix $S_t \in \mathbb{R}^{L_{\text{text}} \times d_t}$ for each time step $t$. Model (Appendix. C.3) and prompt details (Appendix. B) are in appendix.

**Horizon-specific ICL tasks:** ICL-based methods typically utilize a context set to establish reference patterns of feature-label relationships, enabling target prediction by contrasting query features against the context to identify mathematically similar representations. To align this paradigm with multimodal forecasting, we reformulate the horizon-$H$ prediction problem into $H$ horizon-specific conditional sub-tasks. For each $h \in \{1, \ldots, H\}$, we align historical context with future targets using a temporal offset $h$, defining the context set as $\{(\mathbf{c}_t, S_t, y_{t+h})\}_{t=1}^{T-h}$ and the query as $(\mathbf{c}_T, S_T)$. By iterating this process, the model independently generates each point to construct the full sequence $\hat{\mathbf{y}}_{T+1:T+H}$.

### 2.3. Text Attention Guidance Module

Transformer-based ICL models predict targets by attending from queries to context examples. In a numerical-only forecasting backbone, the attention weight vector $\mathbf{a}_{\text{num}}^{(i,j)} \in \mathbb{R}^{1 \times (T-h)}$ for the transformer layer $i$ and the cross-attention head $j$ is:

$$\mathbf{a}_{\text{num}}^{(i,j)} = \text{Softmax}\left(\frac{\mathbf{q}_{\text{num}}^{(i,j)}(K_{\text{num}}^{(i,j)})^\top}{\sqrt{d_{\text{num}}}}\right) \tag{2}$$

where $\mathbf{q}_{\text{num}}^{(i,j)}$ is the transformation of query at step $T$ and $K_{\text{num}}^{(i,j)}$ is the transformation of historical context matrix (up to $T-h$) used to predict $y_{T+h}$. Here, $d_{\text{num}}$ denotes the dimensionality of the feature space after projection. The resulting weights aggregate the value matrix $V_{\text{num}}^{(i,j)}$ into the

Figure 1. **Architectural overview of the TAG-ICF framework.** Multimodal inputs are processed via parallel pipelines: The numerical representation $\mathbf{c}_t$ is formed by concatenating lagged numerical covariates ($\mathbf{x}_t$) and targets ($y_t$) with timestamp features ($\boldsymbol{\tau}_t$), while raw text ($\mathbf{s}_t^{\text{raw}}$) is LLM-synthesized into a lagged embedding matrix $S_t$. The TAG Module projects embeddings via $W_{\text{text}}^{(i,j)}$ to compute a query–context relevance vector $\mathbf{a}_{\text{text}}$, averaged across $L_{\text{text}}$ lookback windows. This is fused with the frozen backbone's numerical attention $\mathbf{a}_{\text{num}}^{(i,j)}$ via a trainable attention gating coefficient $\alpha^{(i,j)}$, creating guided attention that prioritizes contextually relevant historical windows. The resulting representation generates stepwise forecasts over $H$ steps. Trainable parameters ($W_{\text{text}}^{(i,j)}, \alpha^{(i,j)}$) and frozen components (LLM, embedding model, ICL backbone) are marked with flame and snowflake symbols, respectively.

attended representation $\mathbf{a}_{\text{num}}^{(i,j)} V_{\text{num}}^{(i,j)}$.

Our goal is to use textual context to guide which historical examples should be emphasized, without directly mixing textual embedding with numerical feature representation. We compute a text-conditioned relevance vector $\mathbf{z}_{\text{text}}^{(i,j)}$ from lagged textual representations. Using a shared projection $W_{\text{text}}^{(i,j)}$, we project each lagged text embedding into a similarity space and average query–context similarities:

$$\mathbf{z}_{\text{text}}^{(i,j)} = \frac{1}{L_{\text{text}}} \sum_{l=1}^{L_{\text{text}}} \frac{(\mathbf{s}_T^{(l)} W_{\text{text}}^{(i,j)})(S_{1:T-h}^{(l)} W_{\text{text}}^{(i,j)})^\top}{\sqrt{d_{\text{text}}}} \quad (3)$$

where $\mathbf{s}_T^{(l)}$ is the query vector and $S_{1:T-h}^{(l)}$, the context matrix for lag $l$. To maintain parameter efficiency, $W_{\text{text}}^{(i,j)}$ is shared across transformer layers $i$. We define the textual attention as $\mathbf{a}_{\text{text}}^{(i,j)} = \text{Softmax}(\mathbf{z}_{\text{text}}^{(i,j)})$. To integrate this into the ICL backbone, we compute a gated sum of textual and numerical attention for layer $i$ and cross attention head $j$:

$$\mathbf{a}_{\text{guided}}^{(i,j)} = (1 - \alpha^{(i,j)})\mathbf{a}_{\text{num}}^{(i,j)} + \alpha^{(i,j)}\mathbf{a}_{\text{text}}^{(i,j)} \quad (4)$$

The trainable parameter $\alpha^{(i,j)} \in [0,1]$ modulates guidance intensity, with $\mathbf{a}_{\text{guided}}^{(i,j)}$ replacing the vanilla attention $\mathbf{a}_{\text{num}}^{(i,j)}$ during value aggregation. Note that the set of guided transformer layers $i$ is treated as a hyperparameter.

## 3. Experiment

### 3.1. Experimental setup

**Datasets:** To evaluate our framework, we utilize nine real-world datasets from TimeMMD (Liu et al., 2024) spanning 2000–2023 across diverse domains. We specifically employ TimeMMD's web search results rather than scheduled reports, as their temporal density aligns with numerical data to provide contextual signals. For more details on dataset construction and numerical covariates, see Liu et al. (2024).

**Evaluation Details:** Datasets are divided into train, validation, and test subsets using an 80/10/10% split. We evaluate models across prediction horizons $H \in \{4, 6, 8\}$ using a rolling-window approach (stride 1), normalizing all data based on training set statistics. Performance is measured via Mean Squared Error (MSE) and Mean Absolute Error (MAE) in the normalized target space, providing a complementary assessment of large-error sensitivity and overall error magnitude.

Detailed information on the experimental setup, including dataset statistics (app. C.1), baseline descriptions (app. C.2), implementation specifics (app. C.3), and hyperparameter configurations (app. C.4) is provided in the Appendix.

### 3.2. Main Results

**Performance Gains over Multimodal Baselines.** Table 1 summarizes the forecasting performance of TAG-ICF relative to state-of-the-art multimodal baselines on the

*Table 1.* Forecasting performance of TAG-ICF compared with several baselines on several datasets for a forecast horizon of 6. Best results are shown in bold and second-best results are underlined.

| Datasets | GPT4MTS (Jia et al., 2024) | | MM-TSFlib (Liu et al., 2024) | | ChatTime (Wang et al., 2025a) | | ConTextTab (Spinaci et al., 2025) | | TAG-ICF (ours) | |
|---|---|---|---|---|---|---|---|---|---|---|
| | MSE | MAE | MSE | MAE | MSE | MAE | MSE | MAE | MSE | MAE |
| Agriculture | 0.797 | 0.781 | 1.000 | 0.809 | 1.650 | 1.118 | 1.111 | 0.807 | **0.613** | **0.594** |
| Climate | 0.215 | 0.351 | **0.187** | 0.334 | 0.298 | 0.418 | 0.232 | 0.353 | 0.215 | **0.328** |
| Economy | 1.026 | 0.812 | 0.895 | 0.756 | 2.440 | 1.330 | 1.000 | 0.789 | **0.630** | **0.661** |
| Energy | 0.103 | 0.221 | **0.095** | 0.211 | 0.365 | 0.406 | 0.148 | 0.258 | 0.127 | 0.192 |
| Environment | 0.435 | 0.464 | 0.538 | 0.568 | 0.497 | 0.502 | 0.389 | 0.451 | **0.346** | **0.408** |
| Health | 0.642 | 0.503 | 0.633 | 0.504 | 0.949 | 0.586 | 0.500 | **0.416** | **0.466** | 0.419 |
| Security | 20.920 | 3.706 | 12.562 | 2.717 | 27.815 | 4.626 | 8.392 | 2.458 | **7.629** | **2.348** |
| Social Good | 0.238 | 0.363 | 0.511 | 0.621 | **0.142** | **0.274** | 0.572 | 0.549 | 0.193 | 0.314 |
| Traffic | 0.376 | 0.470 | 1.507 | 1.078 | 0.604 | 0.620 | 0.721 | 0.588 | **0.273** | **0.422** |
| Average Rank | 2.94 | 3.00 | 3.00 | 3.56 | 4.22 | 4.33 | 3.33 | 2.89 | **1.50** | **1.22** |

*Table 2.* Impact of the Text Module on TAG-ICF Performance (Horizon 6). Bold values indicate the better error for each metric.

| Datasets | MAE | | | MSE | | |
|---|---|---|---|---|---|---|
| | w/o Text | w/ Text | Gain | w/o Text | w/ Text | Gain |
| Agriculture | 0.638 | **0.594** | 6.90% | 0.696 | **0.613** | 11.93% |
| Climate | 0.331 | **0.328** | 0.91% | **0.213** | 0.215 | -0.94% |
| Economy | 0.767 | **0.661** | 13.82% | 1.039 | **0.630** | 39.36% |
| Energy | **0.189** | 0.192 | -1.59% | 0.325 | **0.127** | 60.92% |
| Environment | 0.414 | **0.408** | 1.45% | **0.346** | **0.346** | 0.00% |
| Health | 0.468 | **0.419** | 10.47% | 0.584 | **0.466** | 20.21% |
| Security | 2.517 | **2.348** | 6.71% | 8.500 | **7.629** | 10.25% |
| Social Good | 0.399 | **0.314** | 21.30% | 0.297 | **0.193** | 35.02% |
| Traffic | 0.427 | **0.422** | 1.17% | 0.345 | **0.273** | 20.87% |

Time-MMD benchmark (forecasting horizon H=6). We provide extended results for varying horizons in in Appendix D.1. TAG-ICF demonstrates superior consistency across all benchmarks, achieving the leading average rank (1.50 MSE; 1.22 MAE) and outperforming all competing methods on six datasets in MSE and seven in MAE.

The proposed framework yields substantial gains on several challenging datasets. Compared to the most competitive baseline, TAG-ICF reduces MAE by 23.94% in agriculture, 12.57% in economy, 10.21% in traffic, and 9.53% in environment. On the security benchmark, a particularly challenging domain where baseline MAEs span a wide range (7.63 to 27.8), TAG-ICF achieves a 9.09% improvement in MSE and 4.48% in MAE, further evidencing its robustness. This discrepancy suggests that while baselines may prioritize the minimization of large outliers, TAG-ICF provides more stable and reliable predictions across the entire forecasting distribution.

**Generalization Across domains.** A notable finding is the high performance variability exhibited by existing multimodal forecasting methods across different domains. For example, ChatTime remains competitive only within the social good sector, but undergoes significant performance degradation in the economy and security domains. Similarly, GPT4MTS and MM-TSFlib display inconsistent ranking behavior, suggesting a lack of cross-domain robustness. While TAG-ICF may not lead on every individual metric, it se-

cures the strongest aggregate average rank, demonstrating consistently competitive performance across all datasets.

**Impact of Multimodal Integration.** Table 2 quantifies the value of textual modalities by comparing TAG-ICF against its "no-text" numerical baseline (forecasting horizon H=6). Overall, our framework exhibits consistent gains from incorporating text, achieving positive improvements on seven datasets and securing substantial gains in several challenging domains. Specifically, TAG-ICF improves MAE by 21.30% on social good, 13.82% on economy, 10.47% on health, and 6.90% on agriculture. These gains suggest that TAG-ICF successfully extracts complementary semantic signals that augment the backbone's numerical forecasting capacity. For further analysis, see the Appendix for: varying forecast horizons (D.1), textual semantics and embedding impact (D.3, D.5), lookback window $L_{text}$ (D.4), and LLM-based raw text utility (D.6).

## 4. Conclusion

We propose TAG-ICF, an in-context forecasting framework that leverages temporally aligned textual context to modulate attention within pretrained backbones. By selectively prioritizing relevant historical patterns, TAG-ICF achieved superior average ranks (1.50 MSE, 1.22 MAE) on the Time-MMD benchmark, demonstrating that attention-level guidance offers a robust alternative to utilize textual context. While the framework remains sensitive to textual granularity and the inherent strength of the underlying ICL backbone, these limitations provide a roadmap for future work. We also intend to transition from global gating to instance-aware mechanisms that dynamically adapt to input-dependent regimes, while extending our module to a broader range of ICL architectures and multimodal tabular learning settings. Ultimately, this work establishes a scalable path for integrating unstructured semantic context into structured learners to improve resilience in complex, real-world forecasting environments.

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

In this Appendix, we provide supplementary details to support the main text. We have also additionally provided further empirical results and analysis. The sections are organized as follows:

- Related work and background (Appendix A)

- LLM Prompt for Text Summarization(Appendix B)

- Experimental Details (Appendix C)

- Ablations (Appendix D)

# A. Related work and background

### A.1. Multimodal time series forecasting with text

Existing approaches to time series forecasting with textual context can be broadly grouped into prompting-based methods and fusion-based methods. Prompting-based methods serialize numerical time series and textual context into natural language prompts and rely on pretrained language models to generate forecasts. Representative works include Xie et al. (2023); Yu et al. (2023); Wang et al. (2024), as well as PromptCast (Xue & Salim, 2023), ChatTime (Wang et al., 2025a), and LLMTime (Gruver et al., 2023), which explore applying LLMs to time series forecasting by converting numerical inputs into textual form. More recently, Context is Key (Williams et al., 2025) highlights the importance of textual context when it contains predictive signals. However, prompting-based methods rely on language-token reasoning over numerical sequences, which can be brittle for precise temporal extrapolation and structured numerical prediction (Tan et al., 2024).

Conventional fusion-based methods instead learn modality-specific representations and combine or align them before prediction. One group of methods focuses on aligning time-series representations with language-model or textual embedding spaces, including Time-LLM (Jin et al., 2024), TEST (Sun et al., 2024), InstructTime (Cheng et al., 2025), TimeCMA (Liu et al., 2025), and LLM-PS (Tang et al., 2025). Another group designs dedicated multimodal architectures that fuse temporal and textual features through encoders, cross-attention, decomposition, or other learned interaction modules, such as GPT4MTS (Jia et al., 2024), MM-TSFlib (Liu et al., 2024), TGForecaster (Xu et al., 2024), MoAT (Lee et al., 2024), CTPD (Wang et al., 2025b), TimeCAP (Lee et al., 2025), ContextFormer (Chattopadhyay et al., 2025), and TimeXL (Jiang et al., 2026). These methods can capture richer cross-modal interactions than direct prompting, but typically rely on supervised training of task-specific multimodal architectures and carefully designed alignment or fusion modules. This can be challenging in realistic settings where aligned text–time series datasets are noisy, or domain-specific.

Overall, prior work primarily asks either how to prompt language models with time-series information, or how to fuse textual and temporal representations in supervised multimodal models. In contrast, our work studies how textual context can be incorporated into structured in-context forecasting backbones, enabling text to guide temporal reasoning without relying on LLM prompting or direct mapping of joint representations to numerical signals.

### A.2. Structured in-context learning

In-context learning (ICL) has recently emerged as a promising paradigm for structured prediction, where models adapt to new tasks by conditioning on examples provided at inference time rather than relying on task-specific retraining. In the tabular domain, foundation models such as TabPFN (Hollmann et al., 2023; 2025; Grinsztajn et al., 2025) and its real-data variant (Garg et al., 2025), TabDPT (Ma et al., 2025), and TabICL (Qu et al., 2025; 2026) have demonstrated strong performance by learning to infer predictive patterns from contextual examples. This makes structured ICL particularly attractive for low-data settings, where training a task-specific model can be unreliable.

Recent work has begun extending this paradigm beyond standard tabular prediction. TabPFN-TS (Hoo et al., 2025) adapts TabPFN-style in-context learning to time series forecasting, showing that structured ICL can be effective for sequential prediction. In parallel, ContextTab (Spinaci et al., 2025) extends tabular ICL to multimodal tabular settings by incorporating textual information alongside tabular features. The versatility of tabular foundation models is further demonstrated by their ability to perform diverse tasks such as causal prediction (Balazadeh et al., 2025; Robertson et al., 2025), generation (Ma et al., 2023; van Breugel et al., 2024), and distillation (Ma et al., 2024; Feuer et al., 2024). Together, these works suggest that ICL-based structured prediction can support both temporal forecasting and text-enhanced prediction.

## B. LLM Prompt for Text Summarization

```
You are a {domain} expert specializing in {target}.

Given a set of related news and articles from the same {week/month}, analyze the
information to infer the likely impact on {target} over the next 1 to 7 {weeks/months}.
Focus on multi-{week/month} drivers.

Instructions:

  • Provide a high-level analysis or insight based strictly on the provided articles,
    including a directional {target} forecast if the data supports a confident
    outlook.

  • Limit your response to a maximum of two sentences.

  • If specific numbers are necessary for the analysis, generalize highly specific,
    granular numbers, e.g., use "roughly 3%" instead of "3.12%", to maintain focus on
    broader trends.

  • STRICTLY AVOID extraneous introductory phrases, e.g., "The articles state," "Based
    on the news". Start your explanation directly.

Articles:  {articles}
```

## C. Experiment Details

This section details the experimental setup used to evaluate the TAG-ICF framework. We first describe more in detail, the nine datasets from the Time-MMD benchmark in Appendix C.1, followed by an overview of the prompting and fusion-based baselines in Appendix C.2. Technical implementation details, including the ICL backbone and textual processing pipeline, are provided in Appendix C.3, with hyperparameter configurations listed in Appendix C.4.

### C.1. Dataset Statistics and Details

We conduct our experiments on nine datasets from the Time-MMD benchmark (Liu et al., 2024), covering domains including agriculture, climate, economy, energy, environment, health, security, social good, and traffic. All datasets are filtered to the period 2000–2023 to ensure temporal consistency across numerical and textual modalities. The environment dataset was originally available at a daily frequency, we aggregated it to weekly frequency to match the temporal granularity of the textual data. Table 3 summarizes the statistical characteristics of the datasets. The benchmark exhibits substantial heterogeneity in temporal granularity, and sequence length. Specifically, the datasets range from low-frequency monthly series (e.g., security, social good, and traffic) to higher-frequency weekly datasets (e.g., climate and health). Such variability introduces diverse forecasting challenges, including non-stationarity, varying seasonal patterns, long-range temporal dependencies, and domain-specific noise characteristics.

*Table 3.* Summary of Time-MMD datasets (Liu et al., 2024). Dimension refers to the number of variables in each dataset.

| Dataset Name/Domain | Dimension | Frequency | Number of Samples |
|---|---|---|---|
| Agriculture | 2 | Monthly | 288 |
| Climate | 9 | Weekly | 1252 |
| Economy | 3 | Monthly | 288 |
| Energy | 8 | Weekly | 1479 |
| Environment | 6 | Weekly | 1239 |
| Health | 15 | Weekly | 1251 |
| Security | 1 | Monthly | 288 |
| Social Good | 1 | Monthly | 288 |
| Traffic | 1 | Monthly | 288 |

Representative time series from each domain are visualized in Figure 2. Beyond differences in temporal resolution, the datasets also exhibit substantial variability in volatility, temporal sparsity, and cross-modal relevance between textual

and numerical signals. The benchmark contains diverse temporal dynamics, including long-term trends (agriculture and traffic), strong seasonal and periodic behavior (traffic and health), regime shifts and non-stationarity (economy and energy), high-frequency fluctuations (environment), and sparse extreme events with heavy-tailed behavior (security). Consequently, these datasets span a broad spectrum of forecasting regimes, ranging from smooth trend-dominated processes to highly volatile event-driven dynamics, making it a challenging benchmark for robust multimodal forecasting.

Time-MMD provides separate numerical and textual datasets for each domain, where the two modalities may exhibit different temporal granularities and coverage periods. To integrate these modalities, each numerical forecasting window is aligned with temporally relevant textual information using an interval-overlap strategy. Specifically, a textual record is associated with a numerical time window if its temporal intervals overlap:

$$\text{text.start\_date} \leq \text{ts.end\_date} \quad \wedge \quad \text{text.end\_date} \geq \text{ts.start\_date}.$$

This alignment strategy enables the incorporation of temporally relevant contextual information while preserving chronological consistency and preventing future information leakage.

### C.2. Baselines

We compare our method against representatives from two main families of multimodal time series forecasting approaches as discussed in Section A: prompting-based approaches and traditional fusion-based approaches. For prompting-based forecasting, we include ChatTime (Wang et al., 2025a), which adapts pretrained language models to time-series inputs through numerical tokenization and vocabulary expansion. For traditional fusion-based forecasting, we include MM-TSFLib (Liu et al., 2024), GPT4MTS (Jia et al., 2024), and ConTextTab (Spinaci et al., 2025). MM-TSFLib independently encodes numerical and textual modalities and combines their respective predictions through a linear weighting mechanism; GPT4MTS adopts an encoder-decoder framework where inputs are first processed by modality-specific encoders before being passed to a GPT-2 decoder for multimodal fusion and final forecasting; and ConTextTab incorporates textual embeddings into a structured ICL model, providing a text-aware ICL comparison. To ensure a fair comparison, all baselines were fitted on the training data, with hyperparameter optimization conducted via random search across a comprehensive range of model-specific parameters.

Note that ChatTime, a prompting-based forecasting model, typically aggregates textual metadata for an entire series rather than individual timestamps. To adapt it to our setting, we strategically selected textual context from the most recent historical windows; this ensures that the model's forecasting is guided by information temporally closest to the target horizon. While this may be suboptimal as ChatTime was not natively trained in this specific configuration, it nonetheless provides a more consistent evaluation framework and ensures the model is provided with the most relevant contextual signals for the task at hand.

### C.3. Implementation details

In this section, we provide a comprehensive overview of the technical configuration of our framework.

**ICL Backbone and Architectural Configuration.** We utilize TabDPT (Ma et al., 2025) as the core ICL backbone. TabDPT is a Tabular Foundation Model designed to generalize across diverse, unseen datasets without task-specific fine-tuning, having been pre-trained on a broad collection of 123 real-world datasets encompassing 32M rows and 2B cells. The model architecture employs a Transformer encoder that treats each table row as a single token, a design choice that ensures row-order invariance and high computational efficiency. Our specific configuration consists of $B = 16$ layers with a 768-dimensional embedding space; each layer includes multi-head self-attention with 8 heads and a feedforward network with an expansion factor of 2. A critical feature of this architecture is the cross-attention split, which permits query points to attend to all context points while restricting context points to attend only to one another. We apply our textual guidance module exclusively to these cross-attention heads to intervene on the interaction between query and context, leaving the self-attention of context tokens intact. The pipeline concludes with a frozen MLP regression output head to generate our final predictions.

**Textual Processing and Embedding Pipeline.** Our textual components are processed through a multi-stage pipeline. We first utilize GPT-5.2 to analyse the raw text data, extracting high-level insights grounded in the facts provided by the textual input. For vector representation, we employ Qwen3-Embedding-8B, a model trained with Matryoshka Representation Learning. This allows us to balance semantic richness with computational efficiency by extracting the first $d_t = 1024$ dimensions from the original 4096-dimensional embeddings.

**Attention Gating Mechanism and Optimization Strategy.** Before computing the text-guided attention scores, the text embeddings are first linearly projected through $W_{text}^{i,j}$ in eq.(3) into head-specific subspaces. Per-head LayerNorm is then applied separately to the projected query and key representations, which helps stabilize their scale within each head during training. To integrate textual guidance while maintaining numerical stability, we reparameterize the attention gating parameter $\alpha_{ij}$ as $\sigma(o_{ij})$, where $\sigma(\cdot)$ is the sigmoid function and $o_{ij}$ is a randomly initialized latent parameter in the logit space. To prevent the gate from becoming saturated at the boundaries $\{0, 1\}$, we enforce a clamping constraint on $o_{ij}$ within the range $[-5, 5]$. We employ a decoupled learning rate strategy, using a specific rate $\eta_{\text{gate}}$ for the attention gating parameters to coordinate their convergence with the pre-trained weights of the backbone. Both the global learning rate and $\eta_{\text{gate}}$ are treated as hyperparameters.

### C.4. Hyperparameters

We list all the hyperparameters used by our pipeline. We conduct random search over the defined ranges for hyperparameter optimization.

*Table 4.* Sumamary of used Hyperparameter settings

| Hyperparameter | Description | Value or Choices |
|---|---|---|
| criterion | The criterion for calculating loss | MAE |
| optimizer | The optimizer | AdamWScheduleFree |
| tune_batch_size | Batch sizes used for fine-tuning | $\{16, 32, 64, 128, 256\}$ |
| train_epochs | Maximal number of training epochs | 50 |
| patience | Early stopping patience | 5 |
| text embedding lookback window $(L_{\text{text}})$ | Textual data lookback window | 3 |
| text_attn_layers $(i)$ | Text attention layer sets | $\{[1, 2], [13, 14, 15, 16], [15, 16], [16]\}$ |
| gate_lr $(\eta_{\text{gate}})$ | Learning rate for gate parameters | $\{0.01, 0.03, 0.1, 0.3\}$ |
| text_attn_lr | Learning rate for text attention layers | $\{0.0003, 0.001, 0.003, 0.01\}$ |
| max_context | Maximum context length | dataset length |
| dropout | Dropout rate | 0.1 |
| gate_logit_clamp | Clamp value for gate logits | 5.0 |
| text_attention_logit_l2 | L2 regularization on text attention logits | 0.001 |
| use_flash | Whether to use FlashAttention | True |

### C.5. Code and Reproducibility

To support reproducibility, the complete codebase and datasets will be publicly released upon acceptance.

### C.6. Hardware and Compute Environment.

All computational experiments were executed on a system equipped with two NVIDIA RTX 6000 Ada Generation GPUs with 48 GB of VRAM.

## D. Ablations

This section presents extended ablation studies and sensitivity analyses to rigorously evaluate the robustness and individual components of the TAG-ICF framework. We first assess the model's stability across varying temporal scales by evaluating performance at forecast horizons of 4 and 8 in Appendix D.1. To isolate the contribution of the textual modality, Appendix D.2 compares our full architecture against a text-less baseline across all datasets. Furthermore, to verify that the framework captures meaningful textual semantics rather than simply benefiting from increased parameter capacity, we conduct text-shuffling and dropping tests in Appendix D.3. We also investigate the model's sensitivity to technical configurations in Appendix D.4 and Appendix D.5, specifically analyzing the impact of the text lookback window and the choice of embedding models. Finally, Appendix D.6 explores the effectiveness of using LLMs to analyze and compress raw textual data prior to forecasting

### D.1. Effect of Different Forecast Horizons

Table 5 and Table 6 report the forecasting performance of TAG-ICF against several recent multimodal forecasting baselines on the Time-MMD benchmark for forecasting horizons of 4 and 8, respectively. Consistent with the results observed for horizon 6, TAG-ICF achieves the strongest and most stable performance across datasets, obtaining the best average rank in both MSE and MAE.

Compared with the best-performing competing baseline, for example, TAG-ICF improves MAE by 31.77% on Agriculture, 27.38% on Traffic for horizon 4. For horizon 8, TAG-ICF improves MAE by 15.67% on Agriculture, 12.85% on Traffic respectively. These results demonstrate that TAG-ICF maintains its advantage across different prediction lengths. Moreover, the consistent performance gains on the same datasets across varying forecasting horizons further indicate the robustness and generalizability of TAG-ICF under different forecasting settings.

*Table 5.* Forecasting performance of TAG-ICF compared with several baselines on several datasets for a forecast horizon of 4. Best results are shown in bold and second-best results are underlined.

| Datasets | GPT4MTS (Jia et al., 2024) | | MM-TSFlib (Liu et al., 2024) | | ChatTime (Wang et al., 2025a) | | ConTextTab (Spinaci et al., 2025) | | TAG-ICF (ours) | |
|---|---|---|---|---|---|---|---|---|---|---|
| | MSE | MAE | MSE | MAE | MSE | MAE | MSE | MAE | MSE | MAE |
| Agriculture | 0.545 | 0.620 | 0.712 | 0.670 | 1.103 | 0.922 | 0.709 | 0.684 | **0.309** | **0.423** |
| Climate | 0.130 | 0.267 | **0.129** | **0.258** | 0.197 | 0.338 | 0.133 | 0.266 | 0.147 | 0.268 |
| Economy | 0.846 | 0.736 | 0.712 | 0.670 | 1.916 | 1.153 | 0.783 | 0.698 | **0.539** | **0.585** |
| Energy | **0.055** | 0.163 | **0.055** | **0.158** | 0.215 | 0.318 | 0.101 | 0.203 | 0.070 | 0.181 |
| Environment | 0.410 | 0.452 | 0.469 | 0.519 | 0.486 | 0.499 | 0.372 | 0.448 | **0.344** | **0.412** |
| Health | 0.435 | 0.414 | 0.453 | 0.433 | 0.712 | 0.482 | 0.323 | **0.321** | **0.304** | **0.321** |
| Security | 20.964 | 3.828 | 11.894 | 2.626 | 27.553 | 4.582 | 8.367 | 2.485 | **7.999** | **2.417** |
| Social Good | 0.186 | 0.316 | 0.764 | 0.763 | 0.158 | 0.283 | 0.346 | 0.374 | **0.138** | **0.268** |
| Traffic | 0.528 | 0.573 | 1.260 | 0.979 | 0.599 | 0.620 | 0.640 | 0.526 | **0.235** | **0.382** |

*Table 6.* Forecasting performance of TAG-ICF compared with several baselines on several datasets for a forecast horizon of 8. Best results are shown in bold and second-best results are underlined.

| Datasets | GPT4MTS (Jia et al., 2024) | | MM-TSFlib (Liu et al., 2024) | | ChatTime (Wang et al., 2025a) | | ConTextTab (Spinaci et al., 2025) | | TAG-ICF (ours) | |
|---|---|---|---|---|---|---|---|---|---|---|
| | MSE | MAE | MSE | MAE | MSE | MAE | MSE | MAE | MSE | MAE |
| Agriculture | 1.309 | 0.951 | 1.411 | 0.972 | 2.361 | 1.340 | 1.784 | 0.969 | **1.228** | **0.802** |
| Climate | 0.303 | 0.421 | 0.269 | 0.385 | 0.414 | 0.495 | 0.375 | 0.445 | **0.262** | **0.376** |
| Economy | 1.187 | 0.897 | 0.985 | 0.789 | 2.729 | 1.444 | 1.274 | 0.898 | **0.976** | **0.757** |
| Energy | 0.148 | 0.267 | **0.138** | **0.259** | 0.524 | 0.483 | 0.202 | 0.306 | 0.190 | 0.308 |
| Environment | 0.503 | 0.499 | 0.596 | 0.614 | 0.496 | 0.496 | **0.402** | 0.456 | 0.435 | **0.451** |
| Health | 0.913 | 0.618 | 0.828 | 0.581 | 1.156 | 0.682 | 0.692 | 0.513 | **0.630** | **0.507** |
| Security | 53.772 | 6.128 | 11.549 | 2.569 | 27.464 | 4.592 | 8.719 | 2.497 | **8.281** | **2.369** |
| Social Good | **0.211** | 0.359 | 0.722 | 0.779 | 0.227 | **0.338** | 0.851 | 0.712 | 0.276 | 0.371 |
| Traffic | 0.552 | 0.537 | 0.914 | 0.816 | 0.589 | 0.604 | 0.713 | 0.613 | **0.327** | **0.468** |

### D.2. Improvement from the Text Module

We evaluate the impact of incorporating textual information in Table 7 (MAE) and Table 8 (MSE).

Table 7 evaluates the contribution of textual information by comparing MAE of each model with and without its text module for a forecasting horizon of 6. Overall, TAG-ICF exhibits the most consistent gains from incorporating text, achieving positive improvements on seven datasets and the strongest average gains on several challenging domains. In particular, TAG-ICF improves MAE by 21.30% on social good, 13.82% on economy, 10.47% on health, and 6.90% on agriculture. These results indicate that the proposed framework effectively uses complementary semantic information from the textual context rather than relying solely on numerical forecasting capacity.

The MSE results in Table 8 are also highly consistent across both metrics: the text module yields substantial MSE reductions across multiple domains, with the most significant gains observed in Energy (60.92%), Economy (39.36%), and Social Good (35.02%). On average, the inclusion of text provides a 21.96% improvement in MSE, an even larger margin than the average MAE gain, confirming that our text-guided module provides robust and significant benefits across evaluation metrics.

A critical observation is that several existing multimodal methods suffer from substantial performance degradation, or negative transfer, when textual context is added. On the MAE results, ChatTime deteriorates by $87.85\%$ on economy and $81.25\%$ on energy, while GPT4MTS and MM-TSFlib also exhibit large negative transfer on datasets such as security and traffic. The MSE results reveal a similar pattern: baseline methods again show pronounced instability when text is incorporated, whereas TAG-ICF maintains robust performance across domains. For example, ChatTime's performance deteriorates by $87.85\%$ on economy and $81.25\%$ on energy, while GPT4MTS and MM-TSFlib exhibit similar instability on the security and traffic datasets. In contrast, TAG-ICF remains remarkably stable across all domains on both MAE and MSE, showing only a marginal reduction on energy MAE $(-1.59\%)$ and Climate MSE (-0.94%) while achieving both MAE and MSE positive gains on eight of the nine datasets. These results suggest that TAG-ICF possesses a superior multimodal learning capability, effectively utilizing informative text while remaining resilient to the less useful text by adaptively learning to down-weight textual attention during training.

*Table 7.* Impact of the Text Module on Forecast Accuracy (MAE) for a forecast horizon of 6

| Datasets | GPT4MTS | | | MM-TSFlib | | | ChatTime | | | ConTextTab | | | TAG-ICF (Ours) | | |
|---|---|---|---|---|---|---|---|---|---|---|---|---|---|---|---|
| | w/o Text | w/ Text | Gain | w/o Text | w/ Text | Gain | w/o Text | w/ Text | Gain | w/o Text | w/ Text | Gain | w/o Text | w/ Text | Gain |
| Agriculture | 0.813 | 0.781 | 3.94% | 0.830 | 0.809 | 2.53% | 0.724 | 1.118 | -54.42% | 0.816 | 0.807 | 1.10% | 0.638 | 0.594 | 6.90% |
| Climate | 0.345 | 0.351 | -1.74% | 0.330 | 0.334 | -1.21% | 0.346 | 0.418 | -20.81% | 0.350 | 0.353 | -0.86% | 0.331 | 0.328 | 0.91% |
| Economy | 0.798 | 0.812 | -1.75% | 0.784 | 0.756 | 3.57% | 0.708 | 1.330 | -87.85% | 0.783 | 0.789 | -0.77% | 0.767 | 0.661 | 13.82% |
| Energy | 0.217 | 0.221 | -1.84% | 0.208 | 0.211 | -1.44% | 0.224 | 0.406 | -81.25% | 0.257 | 0.258 | -0.30% | 0.189 | 0.192 | -1.59% |
| Environment | 0.427 | 0.464 | -8.67% | 0.547 | 0.568 | -3.84% | 0.528 | 0.502 | 4.92% | 0.451 | 0.451 | 0.00% | 0.414 | 0.408 | 1.45% |
| Health | 0.503 | 0.503 | 0.00% | 0.501 | 0.504 | -0.60% | 0.505 | 0.586 | -16.04% | 0.413 | 0.416 | -0.72% | 0.468 | 0.419 | 10.47% |
| Security | 2.900 | 3.706 | -27.79% | 2.137 | 2.717 | -27.14% | 2.698 | 4.626 | -71.39% | 2.443 | 2.458 | -0.61% | 2.517 | 2.348 | 6.71% |
| Social Good | 0.375 | 0.363 | 3.20% | 0.766 | 0.621 | 18.93% | 0.242 | 0.274 | -13.22% | 0.615 | 0.549 | 10.73% | 0.399 | 0.314 | 21.30% |
| Traffic | 0.442 | 0.470 | -6.33% | 1.049 | 1.078 | -2.76% | 0.558 | 0.620 | -11.11% | 0.584 | 0.588 | -0.68% | 0.427 | 0.422 | 1.17% |

*Table 8.* Impact of the Text Module on Forecast Accuracy (MSE) for a forecast horizon of 6

| Datasets | GPT4MTS | | | MM-TSFlib | | | ChatTime | | | ConTextTab | | | TAG-ICF (Ours) | | |
|---|---|---|---|---|---|---|---|---|---|---|---|---|---|---|---|
| | w/o Text | w/ Text | Gain | w/o Text | w/ Text | Gain | w/o Text | w/ Text | Gain | w/o Text | w/ Text | Gain | w/o Text | w/ Text | Gain |
| Agriculture | 0.914 | 0.797 | 12.80% | 1.006 | 1.000 | 0.60% | 0.842 | 1.650 | -95.95% | 1.133 | 1.111 | 1.96% | 0.696 | 0.613 | 11.93% |
| Climate | 0.209 | 0.215 | -2.87% | 0.201 | 0.187 | 6.97% | 0.212 | 0.298 | -40.22% | 0.230 | 0.232 | -0.89% | 0.213 | 0.215 | -0.94% |
| Economy | 0.988 | 1.026 | -3.85% | 0.823 | 0.895 | -8.75% | 0.785 | 2.440 | -210.75% | 0.990 | 1.000 | -1.03% | 1.039 | 0.630 | 39.36% |
| Energy | 0.098 | 0.103 | -5.10% | 0.094 | 0.095 | -1.06% | 0.104 | 0.365 | -252.60% | 0.152 | 0.148 | 2.75% | 0.325 | 0.127 | 60.92% |
| Environment | 0.387 | 0.435 | -12.40% | 0.545 | 0.538 | 1.28% | 0.572 | 0.497 | 13.02% | 0.392 | 0.389 | 0.80% | 0.346 | 0.346 | 0.00% |
| Health | 0.641 | 0.642 | -0.16% | 0.632 | 0.633 | -0.16% | 0.667 | 0.949 | -42.19% | 0.509 | 0.500 | 1.77% | 0.584 | 0.466 | 20.21% |
| Security | 12.891 | 20.920 | -62.28% | 6.670 | 12.562 | -88.34% | 12.180 | 27.815 | -128.36% | 8.293 | 8.392 | -1.19% | 8.500 | 7.629 | 10.25% |
| Social Good | 0.253 | 0.238 | 5.93% | 0.743 | 0.511 | 31.22% | 0.106 | 0.142 | -33.11% | 0.697 | 0.572 | 17.95% | 0.297 | 0.193 | 35.02% |
| Traffic | 0.374 | 0.376 | -0.53% | 1.454 | 1.507 | -3.65% | 0.536 | 0.604 | -12.75% | 0.705 | 0.721 | -2.28% | 0.345 | 0.273 | 20.87% |

### D.3. Evaluating the Significance of Textual Semantics

To investigate whether the performance gains of TAG-ICF stem from meaningful multimodal learning or simply from increased model capacity (parameter count), we conduct two targeted ablation studies inspired by Li et al. (2026): a *text shuffling* test and a *random text dropping* test.

We first evaluate the model's sensitivity to textual content by randomly shuffling the textual contexts across the dataset, thereby decoupling the semantic information from the corresponding numerical time series while keeping the parameter count constant. As shown in Table 9, TAG-ICF outperforms the shuffled baseline by an average of 6.69% (MAE) and 12.64% (MSE) across the three evaluated datasets. Notably, the performance of the shuffled variant is nearly identical to the "No Text" baseline in terms of MAE, with a marginal difference of only 0.06%. This confirms that the model does not achieve gains through parameter scaling alone; rather, it actively utilizes the specific semantic content of the text to modulate attention toward relevant historical examples.

To quantify the model's dependence on textual availability, we systematically drop a percentage of the textual contexts during inference. As reported in Table 10, the model remains stable with negligible changes of $-0.58\%$ (MAE) and $-0.49\%$ (MSE) when 25% of the text is removed, suggesting a degree of robustness to sparse signals. However, accuracy degrades more significantly at a 50% drop rate, where TAG-ICF maintains a 3.53% (MAE) and 5.40% (MSE) lead over the depleted variant. The observed trend, where performance steadily declines as textual guidance is removed, corroborates our hypothesis that

the Text Attention Guidance module effectively leverages textual cues to refine the numerical attention mechanism.

*Table 9.* Impact of Shuffling on Forecast Accuracy

| Datasets | TAG-ICF | | Shuffled | | No Text | |
|---|---|---|---|---|---|---|
| | MAE | MSE | MAE | MSE | MAE | MSE |
| Environment | 0.408 | 0.346 | 0.415 | 0.363 | 0.414 | 0.346 |
| Health | 0.419 | 0.466 | 0.459 | 0.562 | 0.468 | 0.584 |
| Security | 2.348 | 8.090 | 2.555 | 9.093 | 2.517 | 8.500 |

*Table 10.* Impact of the Dropping Text on Forecast Accuracy

| Datasets | TAG-ICF | | Drop 25% | | Drop 50% | | No Text | |
|---|---|---|---|---|---|---|---|---|
| | MAE | MSE | MAE | MSE | MAE | MSE | MAE | MSE |
| Environment | 0.408 | 0.346 | 0.410 | 0.348 | 0.415 | 0.353 | 0.414 | 0.346 |
| Health | 0.419 | 0.466 | 0.423 | 0.475 | 0.430 | 0.484 | 0.468 | 0.584 |
| Security | 2.348 | 8.090 | 2.355 | 7.793 | 2.495 | 8.925 | 2.517 | 8.500 |

**D.4. Impact of Text Lookback Window**

Table 11 evaluates the performance of TAG-ICF on different text lookback windows ($L_{\text{text}}$). Overall, our model demonstrates stable performance across different text lookback window, indicating that performance improvements are achievable in general cases. For Environment, performance improves gradually as the $L_{\text{text}}$ increases from 1 to 5, with MSE decreasing from 0.354 to 0.338 and MAE decreasing from 0.419 to 0.403. Similarly, Security benefits from a longer textual lookback window, where MSE decreases from 8.815 to 7.543 and MAE decreases from 2.537 to 2.316.

For Health, the best result is obtained at $L_{\text{text}} = 3$, while $L_{\text{text}} = 5$ only slightly degrades performance compared with $L_{\text{text}} = 3$. This suggests that although the optimal lookback window may vary across domains, our model still remains robust to different $L_{\text{text}}$ choices. In particular, the performance differences between $L_{\text{text}} = 3$ and $L_{\text{text}} = 5$ are relatively small across the evaluated datasets, showing that the model can effectively leverage historical textual signals without being limited to a specific $L_{\text{text}}$ value.

*Table 11.* Impact of Text Lookback Window on Forecast Accuracy

| Datasets | $L_{\text{text}} = 1$ | | $L_{\text{text}} = 3$ | | $L_{\text{text}} = 5$ | |
|---|---|---|---|---|---|---|
| | MSE | MAE | MSE | MAE | MSE | MAE |
| Environment | 0.354 | 0.419 | 0.346 | 0.408 | 0.338 | 0.403 |
| Health | 0.488 | 0.430 | 0.466 | 0.419 | 0.479 | 0.424 |
| Security | 8.815 | 2.537 | 7.629 | 2.348 | 7.543 | 2.316 |

**D.5. Effect of Text Embedder**

Table 12 shows how different text embedding models influence the forecasting accuracy, but the performance of TAG-ICF does not collapse when replacing Qwen3-8B with the smaller Qwen3-0.6B model. Qwen3-8B achieves lower MAE across all three datasets and lower MSE on Health and Security, suggesting that a larger and more powerful text embedding model can improve the quality of textual signals used by the forecasting model. In the meantime, the performance gap between Qwen3-0.6B is not huge: Qwen3-8B has an average MAE improvement of 3.58% and an average MSE improvement of 2.03%. TAG-ICF still outperforms the baseline without text. This shows that our proposed framework has the generality and flexibility for different embedding models. Overall, these results suggest that TAG-ICF is reasonably robust to embedding model selection, while still benefiting from higher-capacity embedding models in most cases.

*Table 12.* Impact of Text Embedder Selection on Forecast Accuracy

| Datasets | Qwen3-Embedding-0.6B | | Qwen3-Embedding-8B | |
|---|---|---|---|---|
| | MSE | MAE | MSE | MAE |
| Environment | 0.317 | 0.422 | 0.346 | 0.408 |
| Health | 0.501 | 0.433 | 0.466 | 0.419 |
| Security | 8.316 | 2.451 | 7.629 | 2.348 |

## D.6. Effect of LLM Analyzed Textual Data

Table 13 evaluates whether TAG-ICF depends on carefully processed textual inputs or can also benefit from direct embedding from noisy real-world text. LLM-analyzed text are cleaned and processed by GPT-5.2 using the prompt in Appendix B. Using LLM-Analyzed text improves MAE across all three datasets and improves MSE on Health and Security, suggesting that text cleaning and semantic compression can help the model leverage more forecasting-relevant signals. Furthermore, ther performance of our model using raw-text remains competitive, indicating that the performance benefit of the text module does not disappear when using less curated news-style inputs. Overall, these results suggest that TAG-ICF is robust to text quality, while LLM-based analysis provides a modest but consistent advantage on performance.

*Table 13.* Impact of LLM based Analysis of Textual Data

| Datasets | Raw Text | | LLM Analyzed Text | |
|---|---|---|---|---|
| | MSE | MAE | MSE | MAE |
| Environment | 0.335 | 0.430 | 0.346 | 0.408 |
| Health | 0.494 | 0.432 | 0.466 | 0.419 |
| Security | 7.989 | 2.380 | 7.629 | 2.348 |

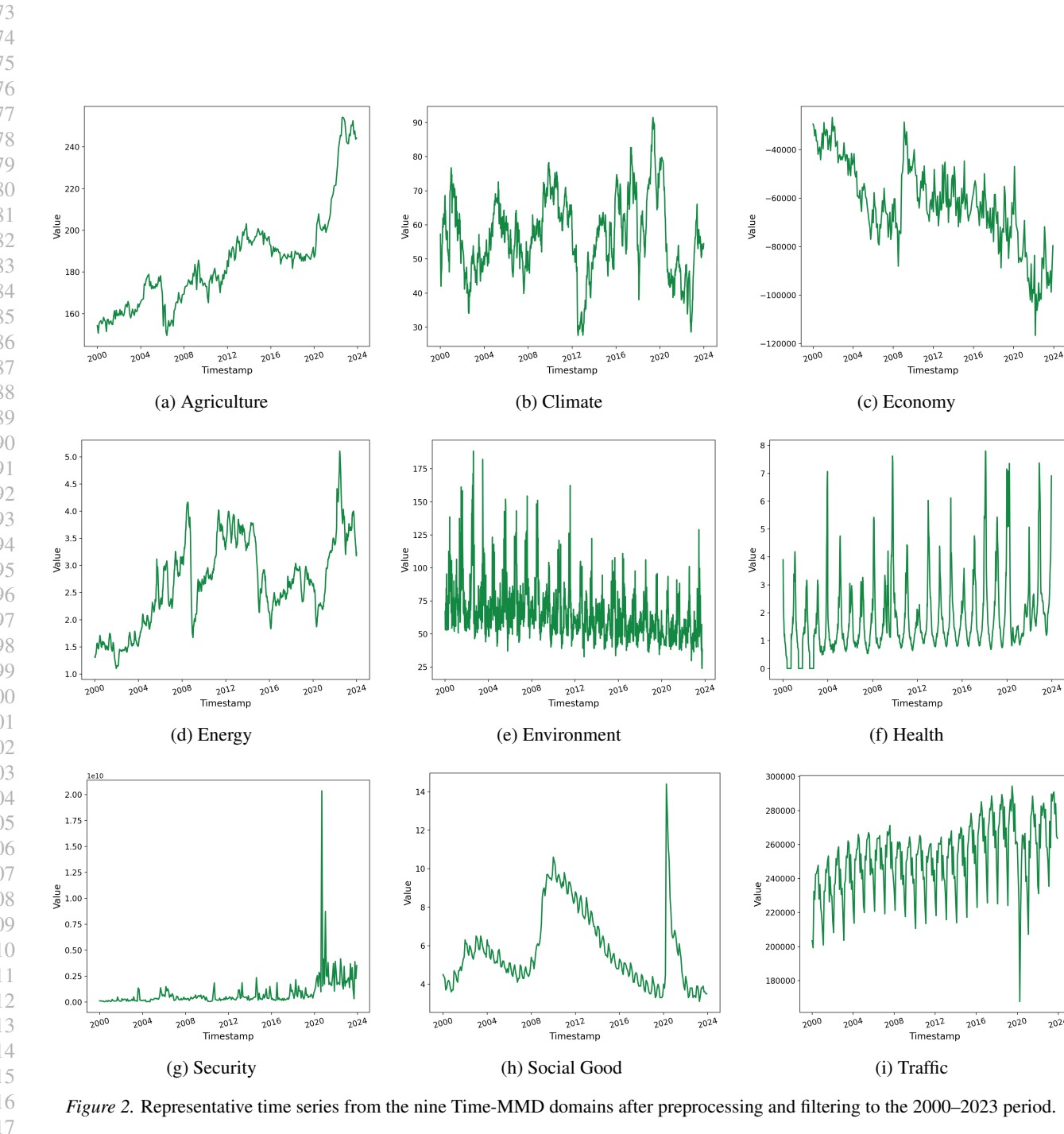

*Figure 2.* Representative time series from the nine Time-MMD domains after preprocessing and filtering to the 2000–2023 period.

