# OpenReview forum: "Text-Attention Guided In-Context Forecasting for Multimodal Time Series Prediction"
_ICML.cc/2026/Workshop/FMSD — FMSD @ ICML 2026 Poster_

### Official Review · Reviewer_Nh3q · 2026-05-20

**Rating:** 7
**Confidence:** 4

**Review:**

# Summary
This paper proposes TAG-ICF, a text-attention guided in-context forecasting framework that improves time-series forecasting by modulating the cross-attention of a frozen tabular ICL backbone (based on TabDPT) using contemporaneous textual information. The method generates textual attention from LLM-based text summaries and embeddings, and combines it with numerical attention through a lightweight gating mechanism. Experimental results on nine Time-MMD datasets across multiple forecasting horizons demonstrate consistent improvements over multimodal baselines and the no-text variant.

# Strengths
- The idea of using text for attention guidance rather than representation-level fusion is novel and elegant.
- The frozen-backbone, parameter-efficient design is practical and modular.
- Empirical evaluation across diverse domains and horizons is reasonably comprehensive and shows consistent gains.
- Ablation studies (e.g., text shuffle/dropout) convincingly support that semantic information contributes to the improvements.

# Weaknesses
- Some notations and implementation details are not very intuitive, making parts of the method slightly difficult to follow on first reading.
- Reproducibility would benefit from clearer descriptions of guided layer/head selections and text alignment details.
- A substantial portion of the performance gain appears to come from the strong tabular backbone itself (as shown by the no-text variant), while the additional contribution of text guidance sometimes seems relatively modest.
- Comparisons against stronger numeric-only ICL baselines such as TabPFN-TS or TabICL-style variants are missing.
- The gating coefficient is implemented as a global scalar, which may limit instance-level adaptivity.

# Overall Assessment
Overall, this paper presents a practical and well-motivated attention-guided ICL framework for multimodal forecasting. The proposed design is modular, parameter-efficient, and technically sound, while the empirical results demonstrate consistent improvements across multiple datasets. The text-guided attention mechanism is a meaningful contribution in the context of structured foundation models. Although clearer reproducibility details and stronger numeric-only baseline comparisons would strengthen the paper further, I believe the work is sufficiently novel and relevant for workshop acceptance.

---

### Official Review · Reviewer_PYZY · 2026-05-20
**Review of Text-Guided In-Context Forecasting for Multimodal Time Series Prediction**

**Rating:** 7
**Confidence:** 4

**Review:**

## Summary
The authors tackle the problem of multi-modal time series forecasting with textual information. Specifically, they propose Text-Guided In-Context Forecasting (TAG-ICF), a text-attention guided in-context learning (ICL) forecasting framework that uses a frozen, pretrained ICL backbone for numerical prediction and uses text to modulate attention over historical examples.

## Strengths
- Good literature review of the multi-modal time series works and the limitations of the current prompting or fusion-based approaches.
- Good application of in-context learning
- Nice motivation of multi-modal time series problem and how the textual information can be beneficial.
- The approach in TAG-ICF differs from other ways to incorporate the textual information including prompting and representation-level fusion.
- Clear problem definition and notation
- Experimental results are strong showing up to 24% gains in MAE and good use of win-rate based metrics as well.
- Nice generalization across wide variety of domains.
- Very detailed appendix

## Areas for Improvement
- Literature is mixed (See cited Zhang et al., 2025) if the additional modalities improves the performance of strong forecasting baselines.
- Only 9 datasets in the evaluation may be limiting. It may also be good to test on the extensive real-world evaluation benchmarks in Zhang et al., 2025.
- The method is tested on TabDPT and it would be interesting to test it on other models as well.
- Ablation studies on the effects of each component in the architecture would be beneficial for improved understanding, e.g., how much benefit is there from the time series adaptation vs. the ICL backbone.
- A major limitation is that Table 1 only compares TAG-ICF to other multi-modal TS models. The table should also include SOTA TSFMs and other strong time series baselines to test if the additional of the textual information is beneficial.
- Missing citation to Chronos-2, which is another ICL-based TSFM.

## Detailed Comments
- Table 1 shows the performance for fixed forecast horizon of 6. It would be interesting to see the performance as a function of the forecast horizon. In particular, how the error accumulates over longer forecast horizons

## Justification of Score
Overall, I find this paper very relevant for the workshop as the time series community is currently actively investigating the best ways to incorporate textual information into time series method. This paper is one of the first to successfully incorporate textual data and provides a novel approach.

---

### Official Review · Reviewer_XXqv · 2026-05-22
**Review of paper 118**

**Rating:** 4
**Confidence:** 4

**Review:**

## Summary
This paper proposes **TAG-ICF** (Text-Attention Guided In-Context Forecasting), a multimodal time-series forecasting framework that leverages a frozen, pre-trained numerical in-context learning (ICL) backbone (specifically TabDPT) and modulates its attention using textual context. Instead of combining text and numerical features through brittle prompting or complex representation-level fusion, TAG-ICF uses an LLM to extract high-level insights from textual context, embeds them, and computes query-context similarity scores. These scores act as an attention gate, weighting historical numerical examples based on their context relevance. Evaluated on nine datasets from the Time-MMD benchmark, TAG-ICF outperforms multimodal and numerical-only baselines, demonstrating robust performance without suffering from the negative transfer issues common in traditional text-fusion methods.

## Pros
- **Modality Isolation to Prevent Semantic Pollution:** By using text strictly to construct attention maps rather than mixing representations, the framework successfully prevents noisy textual embeddings from corrupting the specialized numerical representations of the pre-trained ICL backbone.

## Cons
- **Misleading "Dynamic" Gating Claim:** The authors frame the attention gate as a dynamic modulator. However, the gating parameter $\alpha^{(i,j)}$ is actually a global, static scalar learned per head/layer that does not adapt dynamically to individual input instances at inference time.
- **Severe Bottleneck in Direct Semantic Reasoning:** Because textual representation does not flow into the value aggregation step ($V_{\text{num}}$), the model cannot perform direct semantic reasoning (e.g., interpreting specific numerical changes or directional events mentioned in text). The textual modality serves purely as an instance-routing mechanism, rendering the model entirely dependent on finding historical precedents in the context window.
- **Lossy Uniform Temporal Averaging of Text:** Uniformly averaging text query-context similarities across the lookback window $L_{\text{text}}$ treats older articles with the same weight as recent ones, discarding the chronological sequence and temporal decay of news events.
- **Lack of Dedicated Time-Series Baselines:** The evaluation compares TAG-ICF's backbone against TabDPT (a general tabular foundation model). The lack of comparisons against state-of-the-art specialized time-series foundation models (e.g., Chronos, Lag-Llama, or PatchTST) makes it unclear if the multimodal architecture actually outperforms a strong, dedicated numerical-only forecasting model.